# Parents’ Time Perspective as a Predictor of Child’s Postsurgical Pain, Emergence Delirium, and Parents’ Posttraumatic Stress Disorder Symptoms after Child’s Surgery

**DOI:** 10.3390/children9040539

**Published:** 2022-04-11

**Authors:** Małgorzata Sobol, Marek Krzysztof Sobol

**Affiliations:** 1Department of Psychology, University of Warsaw, ul. Stawki 5/7, 00-183 Warsaw, Poland; 2Hospital Center Châlons-en-Champagne, 51000 Châlons-en-Champagne, France; sobolator@gmail.com

**Keywords:** time perspective, surgery, child pain, emergence delirium, PTSD

## Abstract

Background: The aim of this study was to predict children’s postsurgical pain, emergence delirium and parents’ posttraumatic stress disorder symptoms after a child’s surgery based on the parents’ time perspective. Method: A total of 98 children, aged 2 to 15, and their accompanying parents participated in this study. Measures of parents’ time perspective and posttraumatic stress disorder symptoms were obtained based on questionnaires. The level of children’s postsurgical pain and delirium were rated by nurses and anaesthesiologist. Results: Parents’ future-negative perspective was a predictor of emergence delirium in the group of children aged 8–15 years. Low parents’ past-positive perspective turned out to be a predictor of parents’ posttraumatic stress disorder symptoms after child’s surgery. Conclusions: The results provide evidence for associations between parents’ time perspective with child’s emergence delirium and parents’ posttraumatic stress disorder symptoms after child’s surgery.

## 1. Introduction

A surgery, even a scheduled one, and the hospitalisation connected with it are very strong stressors for children. After waking up from anaesthesia, many children are confused, apprehensive, and under the influence of postsurgical pain. Anxiety and stress may intensify the experience of pain, which in turn leads to an increase in the level of fear and stress [1].

Research into the determinants of postsurgical pain and stress in children is particularly important since the level of pain relief, especially in children, remains unsatisfactory [2,3]. Ineffectively treated postsurgical pain has long-term consequences and leads to numerous complications, such as the development of chronic pain, slower healing of wounds, inflammation conditions, higher proneness to diseases, behaviour disorders, an attitude of distrust towards health service, and may cause higher postsurgical mortality [4]. Moreover, early experience of pain may make the child hypersensitive to pain in the future [5]. Children usually do not understand the short-term nature of pain stimuli [6]. They may perceive pain as a punishment [7] and have limited abilities to cope with pain [8].

Another serious problem is the condition referred to as emergence delirium, postsurgical delirium or emergence agitation, which occurs when the child wakes up from anaesthesia [9]. Emergence delirium is a widespread complication from anaesthesia [10,11]. This condition is described as a mental disorder consisting of hallucinations and disorientation, manifesting itself in loud crying, concern, involuntary physical activity, and thrashing around in bed [11,12]. The rate of postsurgical agitation in children is estimated in various sources at between 2% and 80%. It usually occurs in the first 30 min after waking up from anaesthesia and usually subsides spontaneously, but its duration varies. During a fit of agitation, there is high risk that the child may injure the postsurgical wound or do harm to himself/herself or to a medical staff member [11,12]⁠. The results of research indicate that the occurrence of emergence delirium is related to the choice of anaesthesia, especially with the use of sevoflurane, the child’s age, temper, preoperative anxiety, procedure length, and pain management [11,13]. Prevention and early detection of emergence delirium are very important because of the serious negative consequences this condition has. Emergence delirium in children may result in physical harm, dislocation of catheters, prolonged hospital stay, maladaptive behavioural changes, and dissatisfaction with hospital procedures for parents, children, and staff [9,11].

The high level of anxiety and stress in the parents during medical procedures, their sense of helplessness, and the inability to give their children relief have an impact on how children experience the situation of illness. Researchers indicate positive correlation between the parent’s anxiety level and the child’s distress level after surgery [13]. A parent’s presence near the child during hospitalisation and painful medical procedures is treated as an important strategy in helping the child cope with the difficult situation of treatment [14]. It is not so much a parent’s presence itself during medical procedure that is important as the behaviours the parent engages in with the child—what he or she says and does [14]. It can be said that parents’ behaviour towards children is one of the risk factors for increased distress among children related to the medical procedures they undergo [9]. In various studies, parents’ behaviour explained from about 53 to 64% of variance in the stress experienced by children during short medical procedures [15] and about 50% variance of perioperative stress [16,17,18,19,20]. Researchers emphasise the importance of factors related to parental characteristics in explaining the relationship between their behaviour and children’s behaviour in situations of painful medical procedures [21]. Research results indicate that especially negative affect affects parents’ and children’s behaviour. However, the basics of this mechanism have not been clarified [3].

The situation of a child’s hospitalisation and surgery is also a highly stressful event for the parents [22,23]. Parents are concerned about the health and safety of their child, and they feel they should provide the child with emotional support. After a child’s hospitalisation, parents may suffer from posttraumatic stress disorder symptoms [24,25,26]. Posttraumatic stress disorder symptoms are recognised as recall of negative events, the occurrence of physiological responses to stimuli reminiscent of negative events, a sense of indifference, alienation and of a closed future, difficulty in sleeping and concentrating, as well as irritation and fear [27,28,29]. The results of studies suggest positive correlations between parental anxiety during children’s hospitalisation and parents’ posttraumatic stress disorder symptoms after treatment ends [30].

In the project presented, we treat parents’ time perspective as a variable significant to parents’ influence on the child’s level of postsurgical pain, emergence delirium, and parents’ posttraumatic stress disorder symptoms after surgery. Time perspective is understood as a cognitive style connected with a tendency to focus on a particular area of time: past, present, or future [31]. Six types of time perspective are analysed: past-positive perspective, a tendency to focus on the positively evaluated past; past-negative perspective, a tendency to focus on the negatively evaluated past; future perspective, a tendency to focus on the positively evaluated future; present-fatalistic perspective, a passive attitude, stemming from the belief that life is governed by fate; present-hedonistic perspective, a tendency to focus on pleasure “here and now” [31]; and future-negative perspective, a tendency to focus on the negatively evaluated future [32].

Research results show the considerable significance of time perspective in many areas of functioning [33]. A past-positive perspective was significantly positively associated with a sense of security, received social support, and energy, and was correlated inversely with neuroticism. Focusing on the negatively evaluated past was positively associated with neuroticism, depression, anxiety, problems in social relations, negative mood, and low self-esteem [31,34]. The tendency to focus on the positively evaluated future was significantly positively associated with optimism and health behaviours [31]. Fatalistic and hedonistic perspectives were positively correlated with risky behaviours [35], and fatalism correlated positively with neuroticism [34]. Focusing on the negatively evaluated future was related to past-negative perspective and pessimistic mood [32]. In a study conducted among adult patients undergoing surgery under general anaesthesia, the tendency to focus on the negatively evaluated past was a predictor of the level of postsurgical pain [36]. The results of review analyses of existing studies on the psychological determinants of postsurgical pain in adults pointed to catastrophising as the strongest predictor of acute postsurgical pain [37]. In a study conducted among people with traumatic experiences, there was a significant association between past-negative perspective and fatalistic perspective with posttraumatic stress disorder symptoms [38]. The relationships between parents’ time perspective and the level of postsurgical pain and emergence agitation in children as well as posttraumatic stress disorder symptoms in parents after a child’s surgery have not been investigated yet.

The aim of this study was to test the association between parents’ time perspective [31] with children’s postsurgical pain, emergence delirium and parents’ posttraumatic stress disorder symptoms after the child’s surgery. The following main hypothesis was postulated: there are significant associations between parents’ time perspective with the level of a child’s postsurgical pain, emergence agitation, and parents’ posttraumatic stress disorder symptoms after the child’s surgery. This hypothesis has been formulated based on the biobehavioural model of paediatric pain [39] and the theory of time perspective [31]. The biobehavioural model of paediatric pain posits that in the face of a stressful event such as an operation, cognitive evaluation, coping strategies, and the influence of the family have a significant impact on the child’s experience of pain, their pain-related behaviour, and general functioning [39,40].

We supposed that parents’ tendency to think about negative future scenarios would cause an increase in tension and result in the perception of reality through the lens of danger. Research results suggest that parents with a tendency to exaggerate future threats more often focus on symptoms connected with the child’s experience of pain [20]. Parents’ focus on symptoms connected with a child’s illness and pain in turn intensifies the pain experienced by the child [16,19,41]. In the study by Schechter et al. [19], conducted among parents and children subjected to vaccinations, the strongest predictor of children’s stress behaviours were parents’ predictions regarding the occurrence of those particular behaviours. The research conducted by Caes [42] indicated that parents with a higher tendency to focus on the threat connected with the child’s pain engage in more behaviours connected with pain-related worry (e.g., reassuring) in order to reduce their own stress.

The tendency to focus on the negatively evaluated past is associated with frequent experience of the emotions of grief and sadness and with persistently returning to unpleasant situations from the past. In the research by Sword et al. [38], past-negative perspective correlated positively with Posttraumatic Stress Disorder symptoms [43,44]. In the period after surgery, this can make it more difficult to cope with the memories of difficult moments spent in hospital. Perception of the child’s pain and threat to the child’s life affect the development of posttraumatic stress disorder symptoms in parents [45]. Brown et al. [3] emphasised that persistent psychological distress related to the hospitalisation of the child is caused, among other things, by traumatic memories. For this reason, probably parents with past-negative time perspective would be more likely to experience symptoms of posttraumatic stress.

The specific hypotheses were as follows:

**Hypothesis** **1** **(H1)**.
*Parents’ future-negative time perspective is a predictor of the level of the child’s postoperative pain and emergence delirium after surgery.*


**Hypothesis** **2** **(H2)**.*Parents’ past-negative time perspective is a predictor of the symptoms of parents’ posttraumatic stress disorder symptoms after the child’s surgery*.

## 2. Method

### 2.1. Participants

This research was conducted at the paediatric hospital in Bielsko-Biala, Poland. Children recruited for this study were aged 2–15 years old undergoing outpatient surgery with general anaesthesia. We studied children aged 2–15 to investigate the impact of parental behaviour on both younger and older children. Exclusion criteria included children taking psychiatric medication, developmental delays, and chronic illness. The sample size was determined based on the average effect size in social psychology (r ≈ 0.20). Test power analysis using the G*Power programme [46] was conducted (an average effect f^2^ = 0.15; with high power 0.98). To account for attrition, 100 children and their accompanying parents were enrolled. Two children and their parents were excluded because they had surgery cancelled.

The final sample of participants was 98 children (40 girls and 58 boys; Caucasian 100%) aged 2–15 years (M = 7.77; SD = 3.99) and their accompanying parent. All participants spoke Polish. Children underwent orthopaedic surgery (*n* = 44), urologic procedures (*n* = 24), hernia repairs (*n* = 13), appendectomia (laparoscopy) (*n* = 4), laryngological surgery (*n* = 7), and others (*n* = 6). Participants lived in villages (*n* = 59), small towns (*n* = 15), or medium-sized cities (*n* = 21). Data are available at https://osf.io/8p5kb/, 4 February 2022. Participation was voluntary and remunerated with shopping vouchers. The study protocol and the consent procedure received approval from the ethics committee (Ethic Committee of the Psychology Institute of the University of Wroclaw, consent number IPE 0052, 3 May 2018). The same anaesthesiologist was present during all cases. Nurses (*n* = 3) practising in the hospital were included in this study.

### 2.2. Measures

Parents completed a demographic form concerning the child’s gender and age, the parent’s gender and age, place of residence, and their kinship with the child.

The Zimbardo Time Perspective Inventory (ZTPI [31], adapted into Polish [47] was used to examine parents’ time perspective. This method measures five types of time perspective by means of five scales (56 items with a five-point Likert-type scale—from ‘very untrue of me’ to ‘very true of me’): past-negative (a tendency to focus on the negatively evaluated past; e.g., “Painful past experiences keep being replayed in my mind”); past-positive (a tendency to focus on the positively evaluated past; e.g., “Happy memories of good times spring readily to mind”); future (a tendency to focus on the positively evaluated future; e.g., “When I want to achieve something, I set goals and consider specific means for reaching those goals”); present-hedonistic (a tendency to focus on pleasures “here and now”; e.g., “I try to live my life as fully as possible, one day at a time”); present-fatalistic (a tendency to be passive, stemming from the belief that the future is predetermined, e.g., “Often luck pays off better than hard work”).

The Dark Future Scale (DFS) [32] was used to test the tendency to focus on the negative aspects of the future. It includes 5 items (e.g., “I fear I may have to face crises and difficulties in life”), with a 7-point Likert-type scale (from “very untrue” to “very true”).

The EVENDOL Scale [48]; Polish adaptation [49] was used to measure the level of postsurgical pain in children. It is a behavioural numeric scale. It consists of 5 items: vocal or verbal expression, facial expression, movements, posture, and interaction with the environment. Scores are indicated on 3-point scales. The scale has high internal consistency coefficients (from 0.83 to 0.92) [2,48]. In our study, we used EVENDOL scale as an observational measure (for children aged 2–7) and as a measure of pain based on the child’s interview (for children aged 8–15).

The Paediatric Anaesthesia Emergence Delirium Scale (PAED) [12] in Polish, adaptation by Sobol et al. [50] was used to measure the intensity of postsurgical agitation in the child. The higher the item score, the higher the degree of emergence agitation, i.e., a disturbance in the child’s awareness of and attention to the environment, disorientation, perceptual alterations, hypersensitivity to stimuli, and hyperactive motor behaviour. This scale consists of 5 items, which the person conducting the study rates on a 4-point scale.

The questionnaire PTSD-K [51] was used to evaluate the symptoms of posttraumatic stress in parents after a child’s surgery. The questionnaire consists of 40 items (e.g., “I avoid talking about what happened”). A high score indicates a repeated recall of negative events, the occurrence of physiological responses to stimuli reminiscent of negative events, a sense of indifference, alienation, a sense of a closed future, difficulty in sleeping and concentrating, irritation, and fear.

### 2.3. Procedures

An anaesthesiologist provided parents with information about the psychological research, its objective, procedure, and remuneration for participants, and then asked the parents if they consented to take part in this kind of research together with their child. The parents willing to take part gave their consent in writing. While waiting in the holding area parents completed the following questionnaires: ZTPI and DFS.

Children prepared for anaesthesia were subjected to oral pharmacological premedication with midazolam (Sopodorm, ICN Polfa, Rzeszow, Poland), in a dose of 0.3 mg/kg body mass up to a maximum dose of 7.5 mg, about 30–45 min before anaesthesia—the solution was prepared by the hospital’s pharmacy, based on thyme syrup. Next, all children received intravenous and inhalational general anaesthesia as follows: pre-emptive analgesia—paracetamol (Paracetamol, Fresenius-Kabi, Warsaw, Poland) 15 mg/kg IV up to a dose of 1.0 gr., drip infusion before the beginning of pain impulses from the operative field; induction of anaesthesia—atropine (Atropine Sulfuricum, Polfa, Warsaw, Poland) IV 10 mcg/kg up to a dose of 1.0 mg, dexamethasone (Dexaven, PharmaSwiss, Rzeszow, Poland) IV 0.3 mg/kg up to a dose of 8 mg, fentanyl (Fentanyl WZF, Polfa, Warsaw, Poland) IV 3 mcg/kg, propofol (Plofed 1%, Polfa, Warsaw, Poland) IV 2.5–3 mg/kg, mivacurium (Mivacron, GSK, Brentford, UK) 0.2 mg/kg; maintenance of anaesthesia—sevoflurane (Sevoflurane, Baxter, Deerfield, IL, USA) 1.5–2.0 MAC/N2O/O2, fentanyl 1mcg/kg IV if there were symptoms of insufficient analgesia. PCV ventilation, initially to obtain Vt 10 mL/kg body mass, subsequently ventilation parameters set to maintain normocapnia.

During awakening from anaesthesia, children were extubated after attaining consciousness or at the expiratory concentration of sevoflurane below 0.5% vol. Ten minutes before the expected completion of the procedure, the inflow of sevoflurane was stopped. Next, after attaining an expiratory concentration of the gas below 0.5% vol., the children were ventilated with 100% oxygen to remove nitrous oxide from the breathing mix. After attaining efficient spontaneous breathing and the recovery of consciousness (or at the expiratory concentration of sevoflurane below 0.5% vol.), the children were extubated, and subsequently they were subjected to passive oxygen therapy.

After waking the child from anaesthesia, in the operating theatre, the level of pain was measured at 15 min after waking up, using the EVENDOL scale. The intensity of emergence delirium was measured 15 min after waking up from anaesthesia, using the PAED scale. The last stage of research was conducted 7 days after surgery in the hospital during the control visit when the parent completed the PTSD-K.

## 3. Results

Mean scores, standard deviations and Cronbach’s alphas for the scales are presented in Table 1. In order to examine the relationships between child’s age, parent’s time perspective, child’s postsurgical pain and child’s emergence delirium, child’s age was entered as a continuous moderator and moderator analyses were run for each type of parent’s time perspective. Child’s age was a moderator of the relationship between parent’s future negative perspective and child’s emergence delirium, that is why the analyses were carried out separately for two age groups of children: 2–7 years (*n* = 49; 15 girls and 34 boys) and 8–15 years (*n* = 49; 25 girls and 24 boys). This division was dictated by the fact that children younger than 8 years cannot adequately describe the level of pain they feel due to their level of cognitive and language skills [2,52].

We computed correlations between parent’s time perspective, child’s postsurgical pain, child’s emergence delirium, and parent’s posttraumatic stress disorder symptoms after child’s surgery (Table 2) for a general orientation in the relationships between these variables. In the group of children aged 8–15-years-old, parents’ future-negative perspective correlated significantly positively with a child’s emergence delirium. The results of the linear regression analysis indicate that parent’s future-negative perspective (β = 0.39, *t* = 2.90, R^2^ = 0.14, *p* < 0.01) was a predictor of emergence delirium in this group of children.

A linear regression analysis was conducted to estimate the predictors of parents’ posttraumatic stress disorder symptom intensity. Variables found to be significant in correlation analyses were entered: past-negative, past-positive, and future-negative perspectives (Table 3). The regression model was significant, with 21.4% of the explained variance for the prediction of posttraumatic stress disorder symptoms. It was found that a parent’s weak past-positive perspective turned out to be a predictor of a parent’s posttraumatic stress disorder symptoms.

## 4. Discussion

The aim of the present study was to test hypotheses concerning the associations of parents’ time perspective with a child’s postoperative pain, emergence delirium, and parents’ posttraumatic stress disorder symptoms after the child’s surgery. The findings of our study were partially consistent with the formulated hypotheses. As expected (Hypothesis 1), the parents’ future negative time perspective was a predictor of children’s emergence delirium after recovery from general anaesthesia. The influence of parents’ attitudes, such as exaggerating the threat [20,42] and pessimistic expectations [19], on the level of stress in children experiencing a situation of painful medical procedures has been shown in previous studies. These results can also be compared to the results of examinations of children and their parents in other threatening situations, e.g., floods [27], terrorist attacks [53] or parental trauma experiences [54], which have shown significant relationships between parents’ negative emotional state and stress levels in children.

These relationships can be explained by the fact that children take on the attitude towards time from parents in contact with them in everyday life. The statements about time that they hear from parents shape their way of thinking about the past, present, and future. Presumably, this is particularly strong for older children who already have operational thinking and understand abstract concepts of time [55]. This can explain the lack of significant correlations found in our studies between the time perspective and the level of emergence delirium in the group of younger children. As we reported elsewhere [56], parent behaviour towards their child before surgery was related to the level of child emergence delirium in children aged 2 to 8 years. In turn, the associations of parent behaviour with child emergence delirium were much weaker in the group of older children aged 9–17 years. This suggests that parent behaviour before surgery especially has an influence on a young child’s mental state after surgery. In the case of older children and adolescents, other factors are significant, such as behaviours and attitudes learned by observing the parents’ behaviour in everyday life.

So, probably, among children whose parents have a strong tendency to focus on a negative future in a stressful situation, which in this case is the awakening from general anaesthesia, the perception of this situation is triggered by the parents’ thinking about the threats and the negative future that is to come. In this context, the state of emergence delirium in a child can be viewed as a kind of fear of the threat that is to come. For people with a strong tendency to focus on a negatively perceived future, what is about to happen is primarily seen as a threat [32]. They often exaggerate problems, expecting adverse consequences. This attitude is particularly destructive in situations of experiencing stress related to health problems. A review of the research on the psychological conditions of acute postoperative pain in adults showed that its strongest predictor was catastrophising, i.e., a tendency to exaggerate problems and predict that something terrible will happen in the future [37]. To sum up the results of research so far and the results obtained in the presented study, the impact of future-negative perspective on the perception of pain and distress in situations of medical procedures can be described as the dark future effect.

It is worth noting that in our studies no significant relationship between the parents’ time perspective and the level of postoperative pain in children was found, while there were significant relationships between the parents’ time perspective and personality and emergence delirium in children after surgery. These results suggest that the influence of parents on the child when they experience pain, as assumed by the biobehavioural model of paediatric pain [39,40], is much greater on the child’s behaviour associated with their pain and feelings related to being in a stressful situation than on pain itself.

According to Hypothesis 2, the parents’ time perspective before the child’s surgery had a significant impact on the posttraumatic stress disorder symptoms of the parents after the child’s surgery. However, this is not a tendency to focus on the negative past as anticipated, but low past-positive perspective turned out to be a predictor of posttraumatic stress disorder symptoms. So, it was not so much not returning to unpleasant experiences, but the ability to re-evaluate them and find positive sides to a stressful time that favoured the maintenance of a positive emotional state after returning home from the hospital. The results suggest the need to pay attention to parents with a poor ability to see positive aspects of the past as a risk group for posttraumatic stress disorder symptoms, which requires additional monitoring after the child’s hospitalisation, e.g., during follow-up visits with the child after surgery. Researchers have repeatedly pointed to the adaptive functions of past-positive perspective in dealing with traumatic life experience [31,38,43,44].

The results obtained can be used in practice, especially in dealing with parents of children awaiting surgery. The identification of modifiable psychological variables connected with the intensity of parents’ posttraumatic stress disorder symptoms and child’s emergence delirium has important implications for the development of nonpharmacological interventions that improve postsurgical distress management. Time perspective is a variable of this kind, susceptible to change. It can be formed in a desirable direction through appropriate exercises and programmes [38]. The obtained results can help to answer the question of how to work on parents’ time perspective in such a way as to reduce the level of parents’ posttraumatic stress disorder symptoms after child surgery and emergence delirium in children. The results of the presented study also suggest that in the case of older children undergoing surgery, it may be effective in preventing postoperative delirium to talk with the child about the situation they will find themselves in immediately after waking up from surgery. Explaining to children that this will be a moment of confusion and a new situation for them, but that the condition will pass quickly and they will feel better, can reduce the anxiety of experiencing awakening after general anesthesia.

As regards the study limitations, the results should be carefully generalised to other cultures. A further limitation of this study is the heterogeneity of the paediatric sample. Another limitation is that no data were collected on the parents’ traumatic past experiences. It is interesting idea for further research to analyse the relationships between parent’s difficult experiences from the past and the level of children’s distress during hospitalization. In further studies it would be important to also examine the relevance of cultural aspects in the relationship between parents’ characteristic and behaviour and children’s pain and stress during stressful medical procedures. Researchers emphasise the role of cultural dimension in explaining the expression of pain and distress, the reaction of adults to children’s pain and distress, and the use of strategies to cope [57]. Moreover, it would be worthwhile to check the level of psychological flexibility of the parent as a possible mediator of the relationship between parental time perspective and children’s postoperative distress [58].

Taken together, the results obtained in this study suggest that the parents’ tendency to focus on the negatively perceived future and focusing on threats may significantly intensify the child’s emergence delirium after surgery, which is dangerous for the child and a difficult environment. Moreover, the results of this study found that parents’ past-positive perspective can prevent them from experiencing posttraumatic stress disorder symptoms after a child’s surgery.

## Figures and Tables

**Table 1 children-09-00539-t001:** Descriptive statistics of the parents’ time perspective and posttraumatic stress disorder symptoms after surgery and correlations with child’s pain, emergence agitation and parents’ posttraumatic stress disorder symptoms.

				Correlations		
	M	SD	Cronbach’s Alpha	Child’s Postoperative Pain	Child’s Emergence Delirium	Parents’ Posttraumatic Stres Disorder Sympotms
parents						
gender	-	-	-	−0.02	0.04	−0.02
age	37.11	6.99	-	0.08	0.13	0.01
ZTPI Past-Negative	29.96	6.99	0.81	−0.18	−0.20	0.36 ***
ZTPI Present-Hedonistic	50.31	7.62	0.71	0.17	0.14	0.04
ZTPI Future	48.09	6.17	0.71	−0.04	−0.18	0.08
ZTPI Past-Positive	34.12	4.69	0.62	0.06	0.13	−0.42 ***
ZTPI Present-Fatalistic	24.18	5.43	0.66	−0.11	−0.11	0.12
Future-Negative (Dark Future)	14.75	7.56	0.89	−0.14	−0.21	0.28 **
posttraumatic stress disorder symptoms	19.69	12.94	0.93	0.09	0.01	-
children						
gender	-	-	-	0.07	0.13	−0.02
age	7.77	3.99	-	0.00	0.00	0.05
place of residence’’	-	-	-	−0.08	−0.08	−0.08
postoperative pain	2.18	1.81	0.79	0.00	0.00	0.03
emergence delirium	7.84	2.05	0.83	0.00	0.00	0.01

Note. ** *p* < 0.01; *** *p* < 0.001; M = mean; SD = standard deviation; ZTPI = Zimbardo Time Perspective Inventory; gender was coded: 1 = female, 2 = male; ‘‘ place of residence was coded: 1 = village, 2 = small town; 3 = medium-sized town.

**Table 2 children-09-00539-t002:** Correlations between the parent’s time perspective, child’s postsurgical pain, and child’s emergence agitation.

	Children 2–7 Years Old		Children 2–7 Years Old	
	Child’s Postoperative Pain	Child’s Emergence Delirium	Child’s Postoperative Pain	Child’s Emergence Delirium
parents				
gender	−0.02	0.04	−0.10	−0.25
age	0.08	0.13	−0.10	−0.01
ZTPI Past-Negative	−0.18	−0.20	0.04	0.28
ZTPI Present-Hedonistic	0.17	0.14	−0.04	−0.11
ZTPI Future-Positive	−0.04	−0.18	0.04	0.01
ZTPI Past-Positive	0.06	0.13	0.15	0.01
ZTPI Present-Fatalistic	−0.11	−0.11	−0.08	−0.01
Future-Negative (Dark Future)	−0.14	−0.21	0.13	0.39 **

Note. ** *p* < 0.01; ZTPI = Zimbardo Time Perspective Inventory; ‘ gender was coded: 1 = female, 2 = male.

**Table 3 children-09-00539-t003:** Linear regression results for types of parents’ time perspective as predictors of parents’ posttraumatic stress disorder symptoms (*n* = 98).

Variables	*t*	β	R^2^	∆R^2^	∆F
Model 1			0.21	0.24	8.98 ***
ZTPI Past-Negative	1.73	0.19			
ZTPI Past-Positive	−3.15 **	−0.32			
Future-Negative (Dark Future)	1.34	0.14			

Note. ** *p* < 0.01; *** *p* < 0.001; ZTPI = Zimbardo Time Perspective Inventory.

## Data Availability

We have no previously published or currently in press works stemming from this same dataset. Data are available at https://osf.io/8p5kb/ 4 February 2022.

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
