# Peer review of "Parents’ Time Perspective as a Predictor of Child’s Postsurgical Pain, Emergence Delirium, and Parents’ Posttraumatic Stress Disorder Symptoms after Child’s Surgery"

_children, 2022, doi:10.3390/children9040539_

Round 1
Reviewer 1 Report
This is an important study on a topic that does not receive enough research attention and therefore it is of high importance to publish such articles.
The writers did an impressive job and touched on a fascinating and innovative angle.
I recommend addressing a few points:
- In the introductory part I suggest mentioning the correlation found between the parent's anxiety level and the child's distress level after surgery
Ben Ari, A., Margalit, D., Udassin, R., & Benarroch, F. (2018). Traumatic Stress Among School-Aged Pediatric Surgery Patients and Their Parents. European Journal of Pediatric Surgery 29 (05), 437-442.
- In addition, it is worth mentioning Anne Kazak's study regarding parental perceptions and their effect on a child's PTSD level after surgery.
- Regarding the method - it must be explained why such a wide age range was chosen (2-12). It is known that post-traumatic distress after surgery is expressed differently in preschool children and in school-age children
- In addition, in a study dealing with the effect of the cognitive aspects of the parent on their level of distress after surgery of their child I think it is worthwhile to also check the level of psychological flexibility of the parent (known to be found as a mediating factor affecting children's postoperative distress).
The following is the source: Ben-Ari, A., Aloni, A., Ben-David, S., Benarroch, F. & Margalit, D. (2021). Parental Psychological Flexibility as a Mediating PTSD Factor in Children After Hospitalization or Surgery. The International Journal of Environmental Research and Public Health.
Author Response
Dear Reviewer,
We would like to express our gratitude to you. The review process was really very straight forward and to the point. There were many issues which you pointed out and of course they made it streamlined. I hope we have met the expectations and you will find the implemented changes satisfactory. Thank you for the substantial support.
Yours sincerely
Authors
Responses to the first review
-
In the introductory part I suggest mentioning the correlation found between the parent's anxiety level and the child's distress level after surgery
Ben Ari, A., Margalit, D., Udassin, R., & Benarroch, F. (2018). Traumatic Stress Among School-Aged Pediatric Surgery Patients and Their Parents. European Journal of Pediatric Surgery 29 (05), 437-442.
We mentioned the correlation between the parent’s anxiety and the child’s distress after surgery.
Researchers indicate positive correlation between the parent's anxiety level and the child's distress level after surgery (Ben Ari et al., 2018).
-
In addition, it is worth mentioning Anne Kazak's study regarding parental perceptions and their effect on a child's PTSD level after surgery.
We mentioned Anne Kazak’s study regarding parental and chil’s PTSD.
The situation of a child’s hospitalisation and surgery is also a highly stressful event for the parents (Balluffi et al., 2004; Kazak et al., 2005). Parents are concerned about the health and safety of their child, and they feel they should provide the child with emotional support. After a child’s hospitalisation, parents may suffer from posttraumatic stress disorder symptoms (Sabnis et al., 2019; Shaw et al., 2009; van Warmerdam et al., 2019). Posttraumatic stress disorder symptoms are recognised as recall of negative events, the occurrence of physiological responses to stimuli reminiscent of negative events, a sense of indifference, alienation and of a closed future, difficulty in sleeping and concentrating, as well as irritation and fear (Zawadzki et al., 2002; Cloitre et al., 2019; Gelernter et al., 2019). The results of the studies suggest positive correlations between parental anxiety during children hospitalisation and parents posttraumatic stress disorder symptoms after treatment ends (Best et al., 2021).
-
Regarding the method - it must be explained why such a wide age range was chosen (2-12). It is known that post-traumatic distress after surgery is expressed differently in preschool children and in school-age children
We studied children aged 2-15 to investigate the impact of parental behavior on both younger and older children. Therefore, we divided the study group into two age groups. We added this information to our manuscript.
-
In addition, in a study dealing with the effect of the cognitive aspects of the parent on their level of distress after surgery of their child I think it is worthwhile to also check the level of psychological flexibility of the parent (known to be found as a mediating factor affecting children's postoperative distress).
The following is the source: Ben-Ari, A., Aloni, A., Ben-David, S., Benarroch, F. & Margalit, D. (2021). Parental Psychological Flexibility as a Mediating PTSD Factor in Children After Hospitalization or Surgery. The International Journal of Environmental Research and Public Health.
We mentioned this proposal as a possible direction for further studies. Moreover, it would be worthwhile to check the level of psychological flexibility of the parent a possible mediator of the relationship between parental time perspective and children's postoperative distress (see Ben-Ari et al., 2021).
Reviewer 2 Report
Thank you for the opportunity to review this important contribution to considering the well-being of children and their parents. It considers the parents stance toward expectancies of harm/safety and cognitive attribution styles to the consequences of emergence delirium post-surgery.
The article offers a particularly compelling literature review of the need to explore factors associated with emergence delirium, which can have significant negative consequences. The literature review could be strengthened by a bit of reorganizing. For instance, paragraphs 3 and 4 (lines 36 -55) could be combined - lines 45-46 and 52-55 offer similar information, with lines 52 -55 offering elaboration that strengthens the assertion in lines 45-46. Additionally, paragraphs 5 & 6 seem to be chronologically out of order. I would expect to see discussion of what is stressful about child surgery before a discussion about PTSD after the surgery.
One control I would have wished to see was family/child experience of past surgical procedures and parental history of traumatic or stressful events (such as using the ACEs questionnaire). Was any data collected about parental life experiences or stressors that could be added? The general stance toward attribution of past events (which can then inform future orientation) can be impacted by so many factors - such as parental history of trauma (in which scanning the environment for threat might be adaptive), or cumulative stress from multiple medical procedures for their child. Without more information on parental history, the findings are limited in their generalizability and significance for intervention. If this data was not collected, perhaps this could be addressed as a limitation and future direction? Additionally, did one or both parents complete the questionnaires and were there a different associations based on each parents' responses (i.e., if one parent has a future-positive perspective and one had a future-negative perspective, did that change the association with emergence delirium)?
A reasonable conclusion was drawn as to the difference in findings related to child age, i.e., older children having more exposure and the cognitive capacity to adopt parental appraisal behaviors. However, given that parental affective states are often experienced by young children, I think more discussion of the lack of findings related to young children is warranted. For instance, you note in line 139-141 that pain-related worry changes parental behavior in ways that we might expect to be associated with increased child anxiety, especially for young children who do not know what to expect and thus may be particularly susceptible to parental affect vs. message. I would appreciate hearing more of your thoughts as to the meaning of your lack of findings for young children.
I think the final conclusion speaks to a potent possible intervention - that of priming the parents for positive memories of past events and appreciated the brief nod to intervention. Do the authors have any further thoughts about preventative intervention they could share?
Overall, I appreciated this study and the contribution it makes to the possibility of preventing adverse emotional and physical outcomes post-surgery.
Author Response
Dear Reviewer,
We would like to express our gratitude to you. The review process was really very straight forward and to the point. There were many issues which you pointed out and of course they made it streamlined. I hope we have met the expectations and you will find the implemented changes satisfactory. Thank you for the substantial support.
Yours sincerely
Authors
Responses to the review
The literature review could be strengthened by a bit of reorganizing. For instance, paragraphs 3 and 4 (lines 36 -55) could be combined - lines 45-46 and 52-55 offer similar information, with lines 52 -55 offering elaboration that strengthens the assertion in lines 45-46. Additionally, paragraphs 5 & 6 seem to be chronologically out of order. I would expect to see discussion of what is stressful about child surgery before a discussion about PTSD after the surgery.
We recognized the literature review.
One control I would have wished to see was family/child experience of past surgical procedures and parental history of traumatic or stressful events (such as using the ACEs questionnaire). Was any data collected about parental life experiences or stressors that could be added? The general stance toward attribution of past events (which can then inform future orientation) can be impacted by so many factors - such as parental history of trauma (in which scanning the environment for threat might be adaptive), or cumulative stress from multiple medical procedures for their child. Without more information on parental history, the findings are limited in their generalizability and significance for intervention. If this data was not collected, perhaps this could be addressed as a limitation and future direction?
Thank you for this comment. Data about parental life experiences or stressors were not collected. We mentioned this as a limitation and future direction.
Another limitation is that no data was collected on the parents' traumatic past experiences. It is interesting idea for further research to analyze the relationships between parent’s difficult experiences from the past and the level of children’s distress during hospitalization.
Additionally, did one or both parents complete the questionnaires and were there a different associations based on each parents' responses (i.e., if one parent has a future-positive perspective and one had a future-negative perspective, did that change the association with emergence delirium)?
Only one parent completed the questionnaires.
A reasonable conclusion was drawn as to the difference in findings related to child age, i.e., older children having more exposure and the cognitive capacity to adopt parental appraisal behaviors. However, given that parental affective states are often experienced by young children, I think more discussion of the lack of findings related to young children is warranted. For instance, you note in line 139-141 that pain-related worry changes parental behavior in ways that we might expect to be associated with increased child anxiety, especially for young children who do not know what to expect and thus may be particularly susceptible to parental affect vs. message. I would appreciate hearing more of your thoughts as to the meaning of your lack of findings for young children.
We added more discussion of the lack of findings related to young children.
“This can explain the lack of significant correlations found in our studies between the time perspective and the level of emergence delirium in the group of younger children. As we reported in other place (Sobol et al., 2021), parent behavior towards the child before surgery was related to the level of child emergence delirium in children aged 2 to 8 years. In turn, the associations of parent behavior with child emergence delirium was much weaker in the group of older children aged 9 – 17 years. This suggests that parent behavior before surgery especially has an influence on a young child's mental state after surgery. In the case of older children and adolescents, other factors are significant, such as behaviors and attitudes learned by observing the parents' behavior in everyday life.
So, probably, among children whose parents have a strong tendency to focus on a negative future in a stressful situation, which in this case is the awakening from general anaesthesia, the perception of this situation is triggered by the parents' thinking about the threats and the negative future that is to come. In this context, the state of emergence delirium in a child can be viewed as a kind of fear of the threat that is to come. For people with a strong tendency to focus on a negatively perceived future, what is about to happen is primarily seen as a threat (Zaleski et al., 2019). They often exaggerate problems, expecting adverse consequences. This attitude is particularly destructive in situations of experiencing stress related to health problems. A review of the research on the psychological conditions of acute postoperative pain in adults showed that its strongest predictor was catastrophising, i.e. a tendency to exaggerate problems and predict that something terrible will happen in the future (Sobol-Kwapinska et al., 2016). To sum up the results of research so far and the results obtained in the presented study, the impact of future-negative perspective on the perception of pain and distress in situations of medical procedures can be described as the dark future effect. “
I think the final conclusion speaks to a potent possible intervention - that of priming the parents for positive memories of past events and appreciated the brief nod to intervention. Do the authors have any further thoughts about preventative intervention they could share?
We added a further suggestion of preventive intervention,
The results of the presented study also suggest that in the case of older children undergoing surgery, it may be effective in preventing postoperative delirium to talk with the child about the situation they will find themselves in immediately after waking up from surgery. Explaining to children that this will be a moment of confusion and a new situation for them, but that the condition will pass quickly and they will feel better, can reduce the anxiety of experiencing awakening after general anesthesia.
Overall, I appreciated this study and the contribution it makes to the possibility of preventing adverse emotional and physical outcomes post-surgery.